# STANCE: Motion Coherent Video generation Via Sparse-To-dense ANChored Encoding

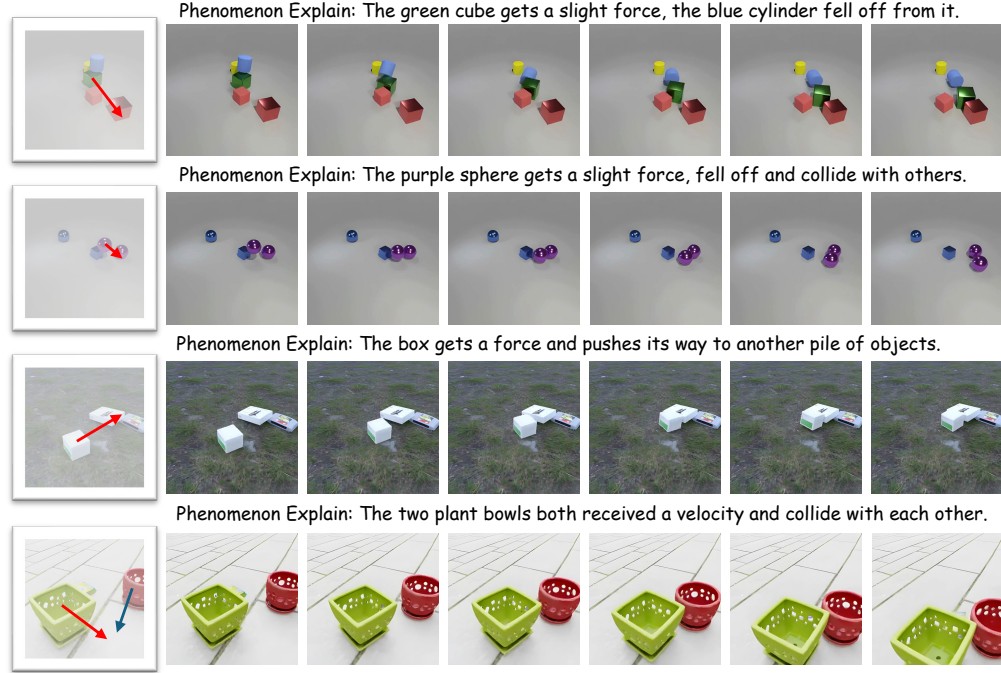

Figure 1: Videos generated by **STANCE**. User input: one keyframe, coarse 2D arrows, per-instance mass, and a scalar depth delta $\Delta z$. Controls yield physically meaningful edits while preserving appearance: increasing mass can reverse collision outcomes, larger speeds produce longer travel and earlier contact, and rotating the arrow reorients trajectories and shifts contact points; $\Delta z$ disambiguates out-of-plane intent under camera motion. Examples span both *simple collision* setups and *realistic scenes*, including gentle pushes that dislodge or trigger collisions.

## ABSTRACT

Video generation has recently made striking visual progress, but maintaining coherent object motion and interactions remains difficult. We trace two practical bottlenecks: (i) human-provided motion hints (e.g., small 2D maps) often collapse to too few effective tokens after encoding, weakening guidance; and (ii) optimizing for appearance and motion in a single head can favor texture over temporal consistency. We present **STANCE**, an image-to-video framework that addresses both issues with two simple components. First, we introduce Instance Cues—a pixel-aligned control signal that turns sparse, user-editable hints into a dense 2.5D (camera-relative) motion field by averaging per-instance flow and augmenting with monocular depth over the instance mask. This reduces depth ambiguity compared to 2D drag/arrow inputs while remaining easy to user. Second, we preserve the salience of these cues in token space with Dense RoPE, which tags a small set of motion tokens (anchored on the first frame) with spatial-addressable rotary embeddings. Paired with joint RGB + auxiliary-map prediction (segmentation or depth), our model anchors structure while RGB handles appearance, stabilizing optimization and improving temporal coherence without requiring per-frame trajectory scripts.

## 1 INTRODUCTION

generation (Chen et al., 2024; Hong et al., 2022; Blattmann et al., 2023b; Yang et al., 2024; Yin et al., 2023) enables synthesizing videos with rich appearance and diverse dynamics for entertainment, XR, driving, and robotics. However, maintaining logical and physical coherence(e.g., consistent trajectories, inertia-like motion, and plausible interactions) remains challenging: models that excel at appearance often still show drift, jitter, or ambiguous contacts, especially when driven by simple control inputs.

One line of work (Liu et al., 2024) tackles coherence by conditioning on full trajectories, which can stabilize object motion but assumes frame-level scripts or dense supervision. In practice, this is rarely available or easy to edit. We focus on a concrete, widely relevant regime: rigid object interactions with contacts, where plausibility depends on getting interaction timing and motion continuity right. These aspects are easy for humans to specify coarsely (e.g., directions, speed hints, relative size/tags) but hard to author frame by frame. Rather than replacing the generative prior with a trajectory controller, we keep the prior in the loop and steer it using sparse, human-editable cues that are lifted into a dense, model-friendly representation.

From a modeling perspective, incoherence does not stem solely from missing "physics." Two pragmatic factors often erode controllability: (i) sparse, low-resolution control maps—especially when injected at a single time slice—can be washed out by tokenization and early attention, leaving too few effective tokens to guide the backbone; (ii) objectives that couple appearance and motion can induce trade-offs, where improving visual quality often comes at the expense of motion consistency. Therefore, a useful control pathway should remain token-dense after encoding, preserve precise spatial alignment, and enable high-quality synthesis while maintaining coherent motion. In our setting, control tokens encode *initial conditions* (who moves, where, how fast, with what mass) rather than full future trajectories, and the DiT is responsible for learning physically reasonable rollouts of rigid-body collisions.

Concretely, we introduce **"Instance Cues"** as a pixel-aligned motion control. During training, we derive per-instance average flow (augmented with monocular depth) and spread it over the instance mask. Unlike 2D drag/arrow interfaces (Zhu et al., 2023; Niu et al., 2024) that lack depth awareness and can be ambiguous under camera motion, our depth-augmented cues encode a direction with depth (2.5D, camera-relative), improving spatial disambiguation. We further introduce a **"Dense RoPE"** mechanism: instead of passing a single low-res map, we select salient spatial locations and assign high-salience, spatial-addressable rotary embeddings to the corresponding motion tokens. Then, we jointly synthesize RGB and a structural stream (segmentation or depth) under the same instance cues. The two streams share spatio-temporal tokens and attend to the same cue tokens, so the structural head acts as a geometry/consistency witness that regularizes the RGB head, tightening alignment and reducing drift without requiring per-frame scripts. Together, these components provide a human-editable description of the initial state, keep it well anchored in token space, and use a structural "witness" to regularize the learned dynamics, aligning the DiT's rollouts with rigid-body intuition.

- **Pixel-aligned, human-editable control.** We introduce a pixel-aligned *2.5D* control interface that turns sparse hints into a dense motion field. The depth-augmented arrows resolve camera-motion ambiguity, remain easy to user.

- **Token-dense injection via Dense RoPE.** We design a Dense RoPE mechanism that selects nonzero sites in each target region and allocates a fixed motion-token budget with *first-frame* RoPE to retain spatial identity. We co-train a lightweight depth head on the same cues as a geometry/consistency "regularizer."

- **Data and validation.** We curate a 200k-clip dataset of rigid-body interactions (single- and multi-object, realistic composites) and run extensive experiments to demonstrate the effectiveness of our method.

## 2 RELATED WORKS

### 2.1 VIDEO DIFFUSION MODELS

Recently, diffusion models have made strong progress on video generation, producing high visual quality with good frame-to-frame consistency. Early works (Blattmann et al., 2023a; He et al., 2022; Wu et al., 2023) mostly adopt UNet-based architectures with encoder–decoder backbones and temporal attention modules (Guo et al., 2023) to enhance temporal coherence. More recently, Diffusion Transformers (DiTs) (Yang et al., 2024; Zheng et al., 2024; Kong et al., 2024; Ma et al., 2025; Wan et al., 2025) have become the dominant choice, as self-attention better captures long-range spatio-temporal dependencies. Compared to UNets, DiTs scale more gracefully to large models and datasets, parallelize more efficiently, and offer a unified architecture for multi-modal conditioning.

### 2.2 MOTION-CONDITIONED VIDEO GENERATION

While recent video diffusion models show impressive visual quality, maintaining coherent motion remains challenging. Unlike pure text-to-video generation—where motion is implicitly induced by abstract prompts—motion-conditioned video generation must accept *explicit* signals that steer dynamics and better align with user intent. Flow-based conditioning has been explored to inject motion cues into the generative process (Shi et al., 2024; Niu et al., 2024; Chen et al., 2023b), and drag-based interfaces let users specify trajectories by placing start/end handles on object parts (Yin et al., 2023; Wu et al., 2024; Li et al., 2025).

Despite this progress, key gaps remain. Control signals are often hard to make and become too sparse after encoding, especially for small or thin objects. Temporal coherence frequently breaks around contacts (hovering, mistimed impacts, "ghost" bounce-backs), and most methods cannot modify object properties such as mass, so user-specified changes in collision outcome (e.g., making the small ball heavier) are rarely respected. Concurrently, VACE (Jiang et al., 2025) proposes an all-in-one DiT-based framework for unified video creation and editing, organizing diverse multimodal inputs (text, frames, masks) into a Video Condition Unit (VCU) and injecting them via a Context Adapter. VACE focuses on broad task coverage, but does not target object-level drag-based control with explicit supervision. Our Instance Cues, Dense RoPE, and joint auxiliary head are complementary to this line of work, aiming instead at a learned, physics-aligned rigid-body simulator under controllable motion editing.

## 3 METHOD

### 3.1 PRELIMINARY

Recent open-source systems adopt diffusion transformers (DiT) for video generation (Yang et al., 2024; Team, 2024). Two design shifts distinguish these models from earlier approaches: *(i)* instead of combining 1D temporal and 2D spatial attention blocks (cerspense, 2023; Chen et al., 2023a; 2024; Zheng et al., 2024), they apply a single 3D spatio-temporal self-attention; *(ii)* text tokens are concatenated with visual tokens and the entire sequence is processed by full self-attention (rather than text-only cross-attention). Full self-attention is then applied across the combined sequence:

$$
\text{Attention}(\mathbf{Q}, \mathbf{K}, \mathbf{V}) = \text{softmax}\left(\frac{\mathbf{Q}\mathbf{K}^T}{\sqrt{d_k}}\right)\mathbf{V}, \quad \text{where}
$$
$$
\mathbf{Z} : \mathbf{Z} \in \{\mathbf{Q}, \mathbf{K}, \mathbf{V}\}
$$
$$
= [\mathbf{W}_{z:z\in\{q,k,v\}}(\mathbf{x}_{\text{text}}); \mathbf{f}_{z:z\in\{q,k,v\}}(\mathbf{x}_{\text{video}})]
$$
(1)

Here $\mathbf{W}_t$ (for $t \in \{q, k, v\}$) represents the projection matrixs in the transformer model, and $\mathbf{f}_t$ (for $t \in \{q, k, v\}$) represents a combined operation that incorporates both the projection and positional encoding for visual tokens. A key modeling choice is the positional encoding for video tokens (indexed by a spatio-temporal position $m$) prior to projection.

There are two commonly used types of positional encoding. One is absolute positional encoding formulated as follows:

$$
\mathbf{f}_{z:z\in\{q,k,v\}}(\mathbf{x}_{\text{video}}) := \mathbf{W}_{z:z\in\{q,k,v\}}(\mathbf{x}_{\text{video}}^m + \mathbf{p}^m),
$$
(2)

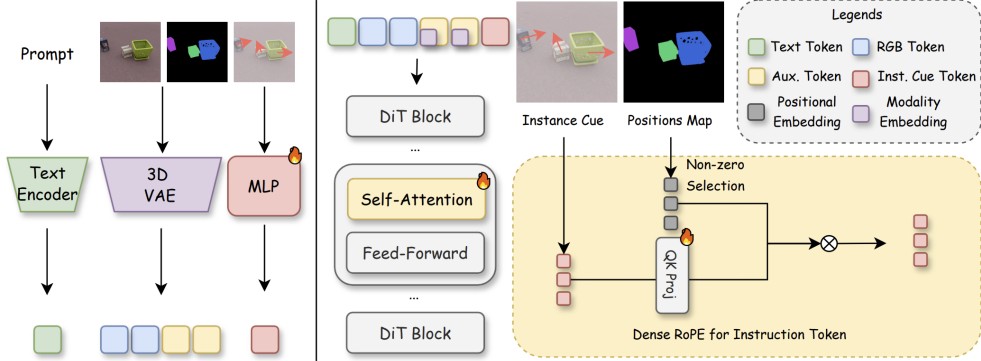

Figure 2: **Pipeline of STANCE.** Our method is organized as follows: (1) **Left:** we extend the input of DiT to include new alpha tokens, and use a train-able MLP to tokenize instance cues. (2) **Right:** The modality embeddings are added to the auxiliary tokens, and the instance cue tokens are paired with Dense RoPE.

where $\mathbf{p}$ is the positional embedding (e.g., a sinusoidal function) and $m$ denotes the position of each RGB video token. Another approach is the Rotary Position Embedding (RoPE) (Su et al., 2024), often used by (Yang et al., 2024; Team, 2024). This is expressed as

$$\mathbf{f}_{z:z\in\{q,k\}}(\mathbf{x}_{\text{video}}) := \mathbf{W}_{z:z\in\{q,k\}}(\mathbf{x}_{\text{video}}^{m}) \circ e^{im\theta}, \tag{3}$$

where $m$ is the positional index, $i$ is the imaginary unit for rotation, and $\theta$ is the rotation angle.

## 3.2 OUR APPROACH

Figure 2 illustrates STANCE's pipeline. Given a text prompt and a keyframe, the user supplies *instance masks*, coarse *arrows* (represents direction and magnitude), and a *mass* tag per instance. We convert these sparse inputs into a dense, pixel-aligned *instance cue* field (2.5D, camera-relative) and inject it into the model with *Dense RoPE* tagging. The model jointly predicts RGB and an auxiliary structural map (segmentation or depth), sharing spatio-temporal tokens and attending to the same cue tokens.

### 3.2.1 SPARSE→DENSE MOTION CUES

We use *instance cues* as the control signal: a few per–object motion hints that are expanded into a dense, mask–aligned field (2.5D when depth is used). It remains easy to use and spatially precise.

**Training.** For each clip we have optical flow $\mathbf{O}$ between two reference frames and an instance map on the first frame. For every instance $i$ with pixel set $\Omega^{(i)}$, we compute a *mean motion vector* by averaging flow over the instance, then *paint* this vector back over renderer-provided ground truth mask to obtain a dense field:

$$\bar{\mathbf{v}}^{(i)} = \frac{1}{|\Omega^{(i)}|} \sum_{(x,y)\in\Omega^{(i)}} \mathbf{O}(x,y).$$

With monocular depth provided, analogous to optical flow, we derive a per-instance *delta depth*: given monocular depths $D_t$ and $D_{t+1}$, we set $\Delta z_i = \text{mean}_{\mathbf{p}\in M_i}\big(D_{t+1}(\mathbf{p}) - D_t(\mathbf{p})\big)$, and append this scalar as the third control channel, yielding a camera-relative "2.5D" cue.

**Inference.** A user specifies a keyframe, per-instance masks (e.g., SAM (Kirillov et al., 2023)), and a coarse 2D arrow for each instance, together with a mass value. We rasterize each arrow inside its mask to obtain a dense in-plane control map. Optionally, the user provides a scalar depth delta $\Delta z$ per arrow, which we broadcast over the same mask and *append as a third control channel* (as in training) to disambiguate motion under camera movement. In Fig. 3, the vertical axis depicts the user-drawn in-plane arrow ($\Delta z = 0$, black); a positive depth delta (red, right of 0) indicates motion into the screen, whereas a negative depth delta (blue, left of 0) indicates motion out of the screen.

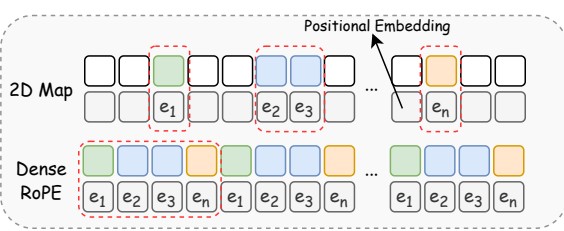 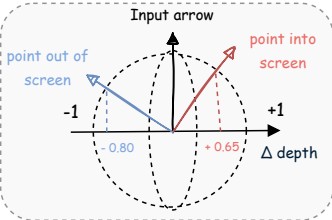

Figure 3: **Left:** *2D Map vs. Dense RoPE.* Downsampling a 2D control map makes many tokens zero, weakening guidance. **Dense RoPE** extracts all nonzero tokens in the target region (colored), assigns positional embeddings $(e_1, \ldots, e_n)$, and feeds a compact dense sequence—yielding stronger, spatially localized control. **Right:** *Depth control.* The user draws a 2D arrow (black). A scalar depth offset $\Delta \in [-1, 1]$ encodes out-of-plane motion: $\Delta > 0$ (red) moves into the screen (away from the camera); $\Delta < 0$ (blue) moves out of the screen (toward the camera).

### 3.2.2 DENSE ROPE

**Motivation:** Downsampling during *patchify* often makes the 2D control mask extremely sparse: a few informative value are surrounded by many zeros, particularly for small or thin region. We keep only the nonzero sites and enforce a fixed token budget, guaranteeing enough motion tokens even for tiny objects. Each selected token is tagged with its first-frame RoPE so its spatial identity persists over time; these tokens act as stable, high-signal anchors that later layers can reliably attend to. Unlike global rescaling, this directly reduces dilution and keeps the control pathway token-dense and spatially aligned after encoding.

Let the first–frame control mask be $\mathbf{M} \in \{0,1\}^L$ on the latent token grid, and let $\mathbf{X}_{\text{Cue}} \in \mathbb{R}^{L \times C}$ denote the per-token control features (e.g., patchified 2.5D instance-cue latents). We collect active indices

$$\Omega = \{i \in \{1, \ldots, L\}: \mathbf{M}_i = 1\}, \qquad m = |\Omega|.$$

To meet a fixed budget $N$ of motion tokens required by the backbone, we form an index list $\mathcal{J}$ by

$$\mathcal{J} = \begin{cases} \text{uniformly subsample } \Omega \text{ to length } N, & m > N, \\ \text{tile and truncate } \Omega \text{ to length } N, & m \leq N, \end{cases} \Rightarrow |\mathcal{J}| = N.$$

Given the selected index set $\mathcal{J}$ and per-token flow features $\mathbf{x}_j^{\text{Cue}} \in \mathbb{R}^C$ for $j \in \mathcal{J}$, we form query $\mathbf{q}$, key $\mathbf{k}$, for the motion tokens using the same $\mathbf{f}_z$ operator as visual tokens, where $\mathbf{p}^j$ is the *first-frame* positional code at site $j$, and scaled by a learnable gain $g$:

$$\mathbf{q}_j^{\text{Cue}} = \mathbf{f}_q(\mathbf{x}_j^{\text{Cue}}) = \mathbf{W}_q(\mathbf{x}_j^{\text{Cue}} + \mathbf{p}_j^{\text{Cue}}), \quad \tilde{\mathbf{k}}_j^{\text{Cue}} = g_k \, \mathbf{f}_k(\mathbf{x}_j^{\text{Cue}}) = g_k \, \mathbf{W}_k(\mathbf{x}_j^{\text{Cue}} + \mathbf{p}_j^{\text{Cue}}), \quad (4)$$

We then concatenate motion tokens into the full sequence. The detailed algorithm flow can be found in the supplementary section.

### 3.2.3 JOINT AUXILIARY GENERATION

**Joint RGB + auxiliary map generation.** We extend the pretrained RGB video backbone to jointly synthesize an *auxiliary* structural stream (segmentation or depth) alongside RGB. Concretely, we duplicate the video token sequence so the model handles two modality-aligned streams of equal length $L$: the first $L$ tokens decode to RGB and the next $L$ tokens decode to the auxiliary map:

$$\mathbf{X}_{\text{video}}^{1:2L} = [\mathbf{X}_{\text{rgb}}^{1:L}; \mathbf{X}_{\text{aux}}^{1:L}].$$

**Positional alignment with a light domain tag.** RGB and auxiliary tokens at the *same* spatio–temporal index share the *same* positional code to enforce pixel/time alignment; a tiny learnable domain embedding marks the additional modality.

$$\mathbf{f}_z(\mathbf{x}^{\text{video}}) = \mathbf{W}_z(\mathbf{x}^{\text{video}} + \mathbf{p}^{\text{video}}), \qquad \mathbf{f}_z^*(\mathbf{x}^{\text{aux}}) = \mathbf{W}_z^*(\mathbf{x}^{\text{aux}} + \mathbf{p}^{\text{aux}} + \mathbf{d}_{\text{aux}}), \quad (5)$$

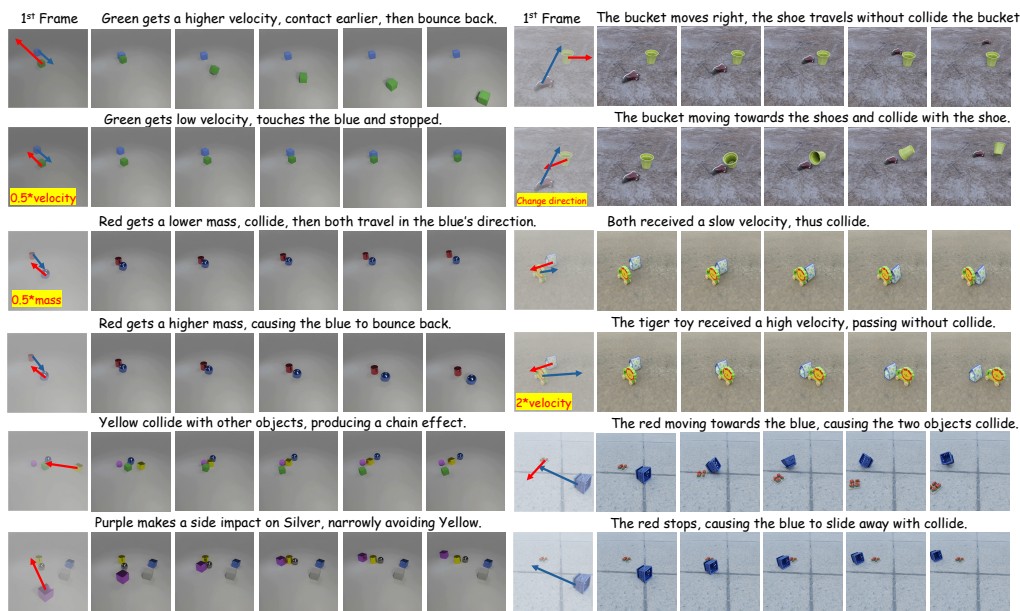

Figure 4: **Applications. Left**: Provide with the first frame, we alter the magnitude of the **velocity** and **mass** for the synthetic objects. **Right**: For realistic objects, the mass is fixed, thus we altered the direction and magnitude of velocity only. **The video clips for all shown cases can be located in the HTML files.**

where $\mathbf{d}_{\text{aux}}$ is a learnable, zero-initialized domain vector; for RoPE we simply apply the *same* rotary index $\theta_m$ to both RGB and auxiliary keys/queries at $m$.

Self-attention is applied over text and both streams:

$$\mathbf{Z} \in \{\mathbf{Q}, \mathbf{K}, \mathbf{V}\} \;=\; \left[\, \mathbf{W}_z(\mathbf{x}_{\text{text}}) \;;\; \mathbf{f}_z(\mathbf{x}_{\text{rgb}}^{1:L}) \;;\; \mathbf{f}_z^*(\mathbf{x}_{\text{aux}}^{1:L}) \,\right]. \tag{6}$$

**Training objective.** We keep the diffusion objective and supervise both heads; the joint loss is

$$\mathcal{L} \;=\; \mathbb{E}_{t,\epsilon}\left[\left\|\hat{\epsilon}_{\text{rgb}} - \epsilon\right\|_2^2 \;+\; \lambda_{\text{aux}}\left\|\hat{\epsilon}_{\text{aux}} - \epsilon\right\|_2^2\right], \tag{7}$$

with a single weighting $\lambda_{\text{aux}}$. Joint prediction stabilizes optimization: the auxiliary stream anchors structure/geometry while RGB focuses on appearance. Empirical comparisons are reported in Sec. 4.2.

## 4 EXPERIMENTS

**Dataset Preparation.** We build our dataset on Kubric, rendering 200k short clips of rigid-body interactions split evenly between (i) simple scenes with single- or multi-ball interactions and (ii) composite realistic scenes with scanned objects and randomized backgrounds. In the simple subset we place one or more rigid objects in a minimal environment and randomize object shape, mass, initial linear velocity, and initial position; lighting uses three rectangular area lights plus a directional sun with fixed placements and randomized intensity. In the composite subset, we replace simple geometry with GSO assets and render against backgrounds sampled from 5,000 environment maps, randomizing object selection, placement, and pose to induce diverse contacts and occlusions. Camera intrinsics/extrinsics and renderer settings are kept consistent within each scene, while material, friction, restitution, and object counts are sampled within bounded ranges to diversify collisions. We construct held-out validation and test splits from both subsets to avoid scene- or asset-level leakage.

**Model.** We fine-tune the **CogVideoX-1.5 (5B) Image-to-Video** backbone with our Instance-Cue injection and Dense RoPE. Unless otherwise noted, the model generates **RGB** videos at $512 \times 512$ resolution, 49 frames, 16 FPS. We finetune for 50k iterations on $8\times$ H100 GPUs; the

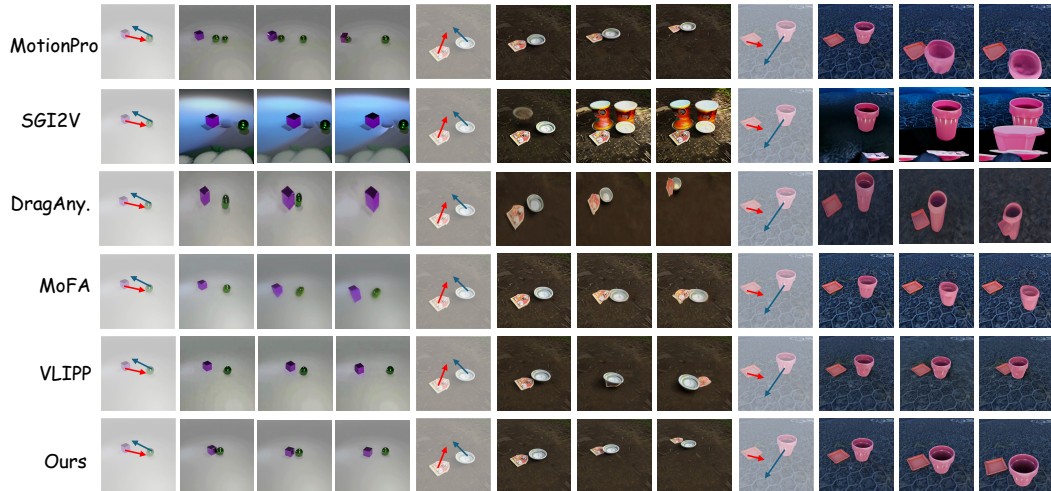

Figure 5: **Qualitative Comparisons.** We compare against recent baselines. All videos are temporally aligned (trimmed or padded to a fixed duration) and spatially normalized (resized for visualization). STANCE attains superior visual fidelity while maintaining strong physical coherence. **The video clips for all shown cases can be located in the HTML files.**

base tokenizer and text encoder remain frozen. For domain conditioning, we use a learnable $1 \times D$ vector (zero-init) that is tiled to $L \times D$ per sequence during training.

**Applications.** Our interface takes a keyframe with instance masks, per–instance arrows that encode the initial velocity $\mathbf{v}_0$, a scalar depth delta $\Delta z$ per arrow, and a per–instance mass scalar $m$. These *Instance Cues* are encoded as our 2.5D (camera–relative) control and injected into the video backbone.

*Speed sweep.* With mass fixed, varying both the magnitude and direction of the initial velocity $\mathbf{v}_0$ yields predictable kinematics: increasing $\|\mathbf{v}_0\|$ increases displacement over a fixed horizon and reduces time-to-contact, while rotating $\mathbf{v}_0/\|\mathbf{v}_0\|$ reorients the trajectory and shifts the contact point; visual appearance remains unchanged. (Fig. 4, top).

*Mass sweep.* With $\mathbf{v}_0$ fixed, changing $m$ alters post–contact behavior: mass variation of the red cylinder reverses the collision outcome—when light, it is deflected by the blue ball; after increasing its mass, it instead pushes through and ejects the ball.(Fig. 4, center).

In multi–object scenes, editing a single arrow or mass produces coherent chain reactions without frame–level scripts (Fig. 4, bottom-left). Thanks to depth–augmented cues and Dense RoPE, guidance remains spatially localized and temporally consistent.

**Evaluation set.** We render an additional **200** held-out clips (100 simple, 100 composite) never seen during training. Evaluation is conducted on two splits: *Regular* scenes and a harder *Small-object* subset (thin/tiny instances). Unless otherwise noted, all methods generate 49 frames at 16FPS and $512 \times 512$ resolution.

**Metric: Physics IQ (↑).** To assess motion coherence, we report **Physics IQ** (Motamed et al., 2025)—a single score that emphasizes *how* things move, not just how they look. Physics IQ aggregates a few simple, normalized checks: (i) *Spatial IoU* (where action happens), (ii) *Spatiotemporal IoU* (where and when action happens), (iii) *Weighted Spatial IoU* (where and how much action happens, weighting by motion magnitude), and (iv) *MSE* (how action happens, penalizing deviation from target motion signals). Scores are combined and rescaled to a 0–100 index (higher is better). Compared to common video metrics (e.g., FVD/LPIPS), Physics IQ is more indicative of *motion coherence* and interaction plausibility, which are the focus of our setting.

Table 1: **Ablations on control and joint supervision.** Evaluated on a held-out set with *Regular* scenes and a harder *Small-object* subset (thin/tiny instances). Rows compare: (i) *text-conditioned* baseline, (ii) *Instance Cues as a low-res 2D map*, (iii) our *Dense RoPE* motion tokens, and (iv) *joint* RGB+Depth/Seg supervision.

| | Physics-IQ ↑ | | FVD ↓ | |
|---|---|---|---|---|
| | Regular | Small | Regular | Small |
| ***Control Signals*** | | | | |
| text-conditioned | 24.08 | — | 97.40 | — |
| *w/* 2D-Map | 43.72 | 31.92 | 56.20 | 58.59 |
| *w/* Dense RoPE | **46.89** | **41.83** | **54.63** | **56.32** |
| ***Joint Aux. Gen*** | | | | |
| Only RGB | 46.89 | 41.83 | 54.63 | 56.32 |
| *w/* Depth | **49.03** | **45.63** | **50.39** | **51.32** |
| *w/* Segmentation | 47.96 | 45.12 | 53.09 | 53.35 |

Table 2: **Baseline comparison.** Models: SG-I2V, Drag-Anything, MoFa-Video, MotionPro, VLIPP, and ours. We report *Physics-IQ* (↑; motion coherence/contact plausibility) and *FVD* (↓; perceptual realism) averaged over a 200-clip held-out set. Best and second-best are highlighted.

| Method | Physics-IQ ↑ | FVD ↓ |
|---|---|---|
| ***Baselines*** | | |
| SG-I2V | 15.42 | 113.54 |
| Drag-Anything | 24.86 | 92.78 |
| MoFA-Video | 29.71 | 98.30 |
| MotionPro | 31.58 | 74.27 |
| VLIPP | 36.40 | 57.90 |
| ***Ours*** | | |
| STANCE-2B | 45.90 | 54.97 |
| STANCE | **47.62** | **50.74** |

### 4.1 COMPARISONS

We compare **STANCE** with strong video baselines and editing-by-control methods, and Fig 5 illustrates the outcomes.

**Control faithfulness.** Given the same keyframe, instance masks, and arrows, **STANCE** adheres to the intended directions and relative magnitudes (speed/mass edits) while preserving object identity across frames. *VLIPP* specify behavior via prompts rather than pixel-aligned cues; this leaves spatial and metric ambiguities (where, how far, how fast), yielding less precise control than STANCE (cf. Fig. 5).

**Contact timing and continuity.** STANCE produces cleaner contact onsets and fewer "hovering" frames before/after impact. While *VLIPP* often achieves strong appearance control, its physical consistency is weaker; e.g., in the synthetic case (Fig. 5, left), two objects begin to bounce back before contact. Conversely, *MOFA*, *MotionPro* provide precise per-object targeting, but tend to struggle with appearance consistency over time (identity/texture drift and mask leakage under longer sequences or viewpoint changes), whereas our joint RGB+structure training under shared instance cues mitigates these failures.

**Quantitative results.** To ensure a fair comparison, we finetune another smaller variant of CogVideoX and report comparisons in Table 2. Across the 200-clip held-out set, our method attains the highest *Physics IQ* (↑), outperforming *SG-I2V* (Namekata et al., 2024), *Drag-Anything* (Wu et al., 2024), *MoFA-Video* (Niu et al., 2024), *MotionPro* (Zhang et al., 2025), and *VLIPP* (Yang et al., 2025).

### 4.2 ABLATION STUDIES

**Dense RoPE.** We keep the backbone, training budget identical and compare: (i) text-only CogVideoX, (ii) CogVideoX + 2D–arrow control (single low-res map), and (iii) ours *with* Dense RoPE. On the full held-out set, Dense RoPE consistently improves *Physics IQ* over all baselines. On the **small-object** subset, gains are most pronounced: when objects occupy few pixels, the 2D map and naïve tokenization collapse to very few effective tokens, leading to weak guidance; tagging a small set of motion tokens with Dense RoPE preserves spatial addressability and reduces drift/identity swaps under occlusion.

**Joint auxiliary head (Seg vs. Depth).** We ablate the auxiliary stream used in joint training: (i) **RGB-only** (no auxiliary map), (ii) **RGB+Seg**, and (iii) **RGB+Depth**. Both joint variants improve *Physics IQ* over RGB-only, indicating that a structural target stabilizes optimization. On the Reg-

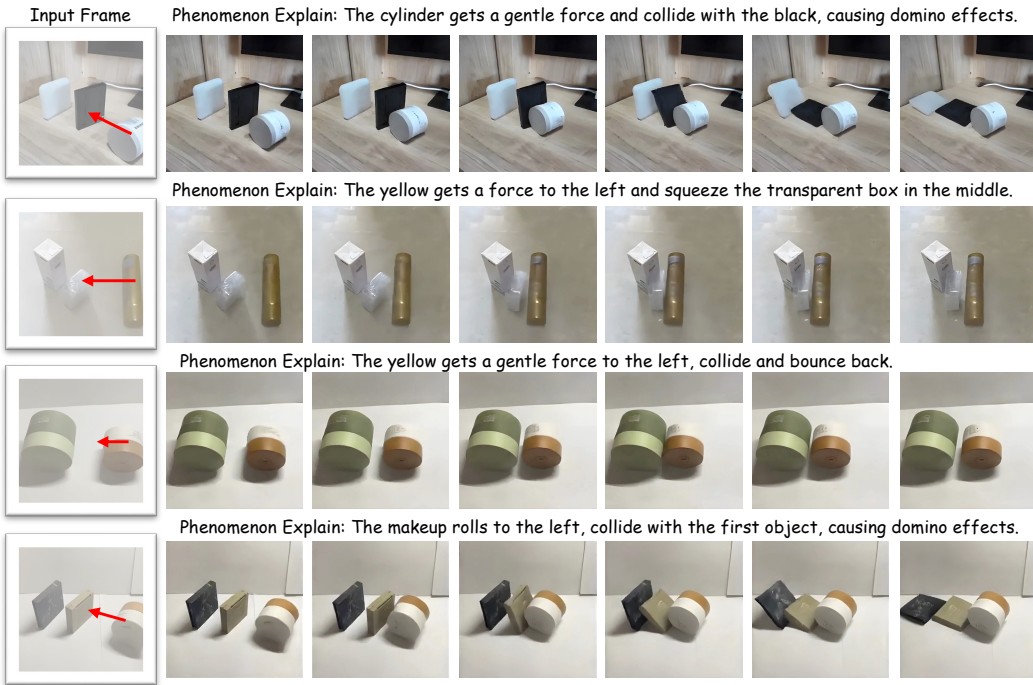

Figure 6: **Real-world demos.** We evaluate **STANCE** on simple tabletop collisions shot with a hand-held phone. Given the first frame and user-specified initial velocities, STANCE follows the directions/speeds, preserves object identity, and produces physically coherent outcomes.

ular split, **depth** yields the highest Physics-IQ: its continuous 2.5D cue improves perspective/order reasoning, making contact/motion to be more coherent than masks alone. On the Small-object split, the gap shrinks as **segmentation's** crisp boundaries give strong spatial anchors when targets are tiny or thin, while monocular depth is noisier at small scales.

### 4.3 REAL-WORLD DEMONSTRATION

**Real-world captured tests.** We evaluate **STANCE** on simple collision scenarios captured with a handheld smartphone (Fig. 6), including tabletop rolling/sliding and two-object contact events. The model follows user-specified directions/speeds and preserves object identity across frames. In a *domino-like chain* example, it maintains consistent appearance for each piece (shape, texture, shading) while producing physically coherent interactions: contacts trigger sequential topples with plausible timing and momentum transfer, without pre-impact "hover" or bounce-back.

## 5 CONCLUSION

We presented **STANCE**, a controllable image-to-video framework that turns sparse, human-editable hints into token-dense, pixel-aligned guidance for coherent motion synthesis. Our *Instance Cues* encode per-instance directions and a camera-relative depth signal ("2.5D") derived during training from flow and monocular depth, which reduces ambiguity under camera motion while remaining easy to author at test time. To keep control effective after encoding, we introduce *Dense RoPE*: instead of a single low-resolution map, we select salient spatial locations and assign spatially address-able rotary embeddings to their motion tokens, preserving alignment and strengthening the control pathway. We further couple RGB generation with a lightweight structural head (segmentation or depth) that attends to the same cues, acting as a geometry/consistency witness and reducing drift without requiring frame-level scripts. Across single- and multi-object settings and realistic scenes, extensive ablations indicate that STANCE improves temporal and interaction coherence while maintaining high visual quality.

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

SUPPLEMENTARY MATERIAL

## A LIMITATIONS

**Limitations and scope.** *Dataset coverage.* Our training set does not include fixed boundaries (e.g., walls, table edges, corners). As a result, interactions that require wall contact and elastic rebound (bounce-back) are not supported: near-boundary motion may lack a realistic impulse response, and glancing edge hits can terminate without a proper reflection. Extending coverage to static boundaries (with normals and restitution/friction parameters) is a straightforward avenue for future work. *Scene/material scope.* Highly non-rigid objects (cloth, ropes, deformables, liquids) are outside our scope. *Depth caveat.* If the user-drawn arrow is nearly frontal (along the camera axis), small depth-scale mismatches can make the motion appear slightly too fast or too slow. Even so, for short everyday shots with modest motion and 1–3 objects, the method produces controllable, coherent results. Looking forward, we are actively working to integrate our components with additional MMDiT-based video backbones to facilitate broader adoption and community benchmarking.

## B BASELINE COMPARISON PROTOCOL

To ensure a fair comparison, we standardize the evaluation as follows.

**Common generation setup.** Unless a baseline mandates otherwise, we use identical sampling budget, resolution, and frame count as in the main paper's evaluation (same prompts and seeds across methods).

**Control adaptation.** For baselines with public training code (e.g., *MoFA-Video* (Niu et al., 2024) and *MotionPro*) (Zhang et al., 2025), we fine-tune the official implementations on our training split using author-recommended settings, matching our evaluation spec (frames, resolution) and without architectural changes. For methods that are inference-only for us (e.g., *SG-I2V* (Namekata et al., 2024), *Drag-Anything* (Wu et al., 2024), *VLIPP*) (Yang et al., 2025), we run their released checkpoints on our validation set. When a baseline natively supports 2D arrow/drag control (e.g., *Drag-Anything*), we provide its native control inputs; otherwise we run the method text-only. For baselines that accept masks, we pass the same first-frame instance masks used by ours. No per-frame trajectories or oracle physics are supplied to any baseline. Outputs are center-cropped/resized and temporally trimmed/padded to the evaluation spec, and we use default or author-recommended inference hyperparameters (steps, guidance) without per-method tuning on the eval set; seeds are fixed where supported.

**Pre/post processing.** All outputs are temporally trimmed or padded to the target length for metric computation. If a baseline generates at a different native resolution, we center-crop and resize with area interpolation before evaluation.

## C DEPTH CONTROL CHANNEL AND RASTERIZATION (SIMPLIFIED)

**Inputs.** For each instance $i$, the user gives (i) a binary mask $M_i$, (ii) a coarse 2D arrow drawn on the keyframe, and (iii) an optional scalar depth delta $\Delta z_i$. Mass $m_i$ is provided in a separate channel and is independent of depth. The sign convention matches Fig. 3: the vertical black arrow means no depth ($\Delta z = 0$); red indicates motion *into* the screen ($\Delta z > 0$, away from the camera); blue indicates motion *out of* the screen ($\Delta z < 0$, toward the camera). The dashed verticals in the figure simply visualize $|\Delta z|$.

**From a user arrow to a dense in-plane control.** Intuitively, we fill the mask with a small vector field that points along the user arrow; vectors are strongest on the drawn line and smoothly fade within the same object, and are zero outside the mask. Concretely, let $\hat{\mathbf{a}}_i$ be the unit direction of the user arrow. For a pixel $\mathbf{p} \in M_i$, we set

$$\mathbf{C}_i^{xy}(\mathbf{p}) = \alpha(\mathbf{p})\, \hat{\mathbf{a}}_i, \quad \alpha(\mathbf{p}) \in [0,1],$$

where $\alpha(\mathbf{p})$ is a soft weight that decreases with the distance from the drawn arrow and is clipped to $[0, 1]$. (Equivalently: draw the arrow as a thin segment inside $M_i$ and apply a small blur restricted to $M_i$; we use a blur radius $\sigma \approx \min(H, W)/20$.)

**Depth channel.**   Depth is a single number per instance, copied to all pixels of the mask and appended as a third control channel:

$$C_i^z(\mathbf{p}) = \Delta z_i \quad \text{for } \mathbf{p} \in M_i, \qquad C_i^z(\mathbf{p}) = 0 \text{ otherwise.}$$

If the user does not specify depth, we default to $\Delta z_i = 0$, i.e., purely in-plane control. This channel helps the model tell apart intended out-of-plane motion from apparent image-plane motion caused by camera parallax.

**Overlapping masks.**   When masks overlap, we take the control from the arrow that is spatially *closest* to the pixel (largest $\alpha$), which yields crisp boundaries in practice. Other simple tie-breakers (e.g., z-order or mass priority) behave similarly; we keep the "closest arrow wins" rule for simplicity.

**How the model sees the control.**   During training we concatenate the control channels $(u, v, \Delta z)$ (and the mass channel, if used) with RGB along the channel dimension. Inference uses the same formatting, so user edits directly map to the inputs the model has seen during training.

**Defaults and ranges.**   We normalize the arrow magnitude to at most 1 after rasterization, and keep $\Delta z$ in $[-1, 1]$. Unless otherwise stated, we set the blur radius to $\sigma \approx \min(H, W)/20$ and do not apply extra scaling ($\lambda = 1$).

# D   DENSE RoPE: ADDITIONAL DETAILS

**Algorithm overview.** We provide a comprehensive, self-contained algorithmic flow for *Dense RoPE* token preparation; see Alg. 1 below.

---

**Algorithm 1** Dense RoPE token preparation

---

**Require:** mask $M \in \{0, 1\}^{B \times h \times w}$, flow features $F \in \mathbb{R}^{B \times C \times H \times W}$, token budget $N$, RoPE bank $(\mathrm{Cos}, \mathrm{Sin})$ aligned to the current image stream

**Ensure:** motion embeddings $T \in \mathbb{R}^{B \times N \times d}$, updated RoPE bank $(\mathrm{Cos}', \mathrm{Sin}')$, sampled indices $\mathcal{J}$

1: **Tokenize:** $M \leftarrow \mathrm{FLATTEN}(M) \in \mathbb{R}^{B \times n}$;  $X \leftarrow \mathrm{PATCHIFY}(F) \in \mathbb{R}^{B \times n \times C'}$        (share token length $n$)

2: **Split RoPE bank:** $\mathrm{Cos}_{\mathrm{base}} \leftarrow \mathrm{Cos}[:-n], \mathrm{Sin}_{\mathrm{base}} \leftarrow \mathrm{Sin}[:-n]$;  $\mathrm{Cos}_{\mathrm{img}} \leftarrow \mathrm{Cos}[-n:], \mathrm{Sin}_{\mathrm{img}} \leftarrow \mathrm{Sin}[-n:]$

3: **for** $b = 1$ **to** $B$ **do**

4:     **Active set:** $I_b \leftarrow \{ i \in \{1, \ldots, n\} \mid M_b[i] = 1 \}$

5:     **Sample indices:**

$$\mathcal{J}_b \leftarrow \begin{cases} \text{uniform sample } N \text{ distinct from } I_b, & |I_b| \geq N, \\ \text{sample with replacement to length } N \text{ from } I_b, & |I_b| < N \end{cases}$$

6:     **Gather:** $X_b^\star \leftarrow X_b[\mathcal{J}_b]$; $\mathrm{Cos}_b^\star \leftarrow \mathrm{Cos}_{\mathrm{img}}[\mathcal{J}_b]$; $\mathrm{Sin}_b^\star \leftarrow \mathrm{Sin}_{\mathrm{img}}[\mathcal{J}_b]$

7: **end for**

8: **Stack:** $X^\star \in \mathbb{R}^{B \times N \times C'}$, $\mathrm{Cos}^\star, \mathrm{Sin}^\star \in \mathbb{R}^{B \times N \times d/2}$

9: **Update RoPE bank:**     $\mathrm{Cos}' \leftarrow \mathrm{CONCATTOKENS}(\mathrm{Cos}_{\mathrm{base}}, \mathrm{Cos}^\star)$;    $\mathrm{Sin}' \leftarrow \mathrm{CONCATTOKENS}(\mathrm{Sin}_{\mathrm{base}}, \mathrm{Sin}^\star)$

10: **Project to model width:** $T \leftarrow \mathrm{FLOWPROJ}(X^\star) \in \mathbb{R}^{B \times N \times d}$

11: **return** $T$, $(\mathrm{Cos}', \mathrm{Sin}')$, $\mathcal{J} = \{\mathcal{J}_b\}_{b=1}^B$

---

# E   VIDEO RESULTS

All video results referenced in the main paper are provided as companion HTML files in the supplementary package. Please open the attached HTML pages with a modern browser to view the clips.

## F    ADDITIONAL VISUAL RESULTS

Beyond the videos shown in the main paper, we include additional generations in the supplementary files, covering diverse objects, motions, and visual effects. Please refer to the attached assets for the complete set of examples.

## G    REPRODUCIBILITY STATEMENT

We will release: (i) training and inference code with pinned dependencies, (ii) pretrained checkpoints for all reported variants (Dense RoPE; joint RGB+Depth / RGB+Seg), (iii) the full Kubric-based dataset used in the paper together with generation scripts, object lists (GSO IDs), and environment-map manifests, and (iv) evaluation code for *Physics-IQ*.

## H    WRITING ASSISTANCE (LLM USE DISCLOSURE)

We used an LLM-based writing assistant (ChatGPT) solely for *language polishing*. The tool helped improve clarity, grammar, and academic tone; it did not generate ideas, analyses, models, figures, or experimental results. All technical content, claims, and conclusions were authored, verified, and approved by us. We reviewed every suggested edit for factual correctness and style, and we remain responsible for the final manuscript. No private or sensitive data were provided to the tool; math expressions, citations, labels, and notation were preserved.

