# OpenReview forum: "STANCE: Motion Coherent Video Generation Via Sparse-To-dense Anchored Encoding"
_ICLR.cc/2026/Conference — ICLR 2026 Conference Desk Rejected Submission_

### Official Review · Reviewer_97AZ · 2025-10-30

**Soundness:** 3
**Presentation:** 3
**Contribution:** 2
**Rating:** 4
**Confidence:** 4

**Summary:**

This paper proposes STANCE, a controllable image-to-video framework that focuses on rigid object interactions and aims to improve motion coherence and interaction plausibility. The method converts sparse 2.5D user cues into dense instance-aligned motion fields, enhances them with a Dense RoPE tagging scheme, and jointly predicts RGB and structural maps to stabilize training. Experiments show improved temporal and physical consistency compared to baselines.

**Strengths:**

1. **Clear problem focus**
   The paper identifies practical challenges in motion-coherent video generation, particularly the loss of control signal density and the difficulty of achieving temporal and physical consistency.

2. **Practical design**
   The sparse-to-dense instance cue formulation is intuitive, human-editable, and easy to integrate into existing video diffusion pipelines.

3. **Reasonable overall design**
   While the components are not entirely novel, the overall framework is coherent and the design choices are appropriate for improving controllability and motion coherence in rigid-object video generation.

**Weaknesses:**

1. **Limitation of Using 2.5D Conditioning**
   The method relies only on 2.5D cues for motion conditioning, while recent approaches such as *Diffusion as Shader* [1] have demonstrated the feasibility of incorporating full 3D conditioning for controlled video generation. Since 3D information can potentially enhance both spatial and physical coherence, which are central to this work, the decision to limit conditioning to 2.5D appears restrictive. Moreover, for example, point-based 3D representations are inherently denser than the 2D control masks discussed in the paper and could naturally alleviate the sparsity issue that the proposed method aims to address. A clearer justification for this design choice would strengthen the paper.

2. **Limitation of Dense RoPE Design**
   Dense RoPE is an effective heuristic for retaining sparse motion cues, but it does not fundamentally resolve the sparsity of the input control signals. The task itself focuses on representing object motion, yet the current RoPE design does not explicitly account for dynamic spatial changes or motion-aware positional encoding. Recent work on dynamic or trajectory-aligned positional embeddings [2] has explored such aspects, and considering this trend, the use of a static first-frame RoPE feels somewhat outdated and not particularly efficient, making it difficult to see clear advantages of this choice.

3. **Limited Novelty in Joint Auxiliary Generation**
   The idea of jointly generating auxiliary structural cues alongside RGB has been explored in prior work (e.g., [3], [4]) and is not particularly novel. While this strategy can help stabilize training and improve motion coherence, similar multi-head or multi-stream supervision schemes have been widely adopted in recent video generation and 3D-aware diffusion models. Therefore, it is difficult to attribute clear novelty to this component. If the authors intend to highlight this as a contribution, it would be helpful to cover relevant prior work in the related works section and clarify how their formulation differs from existing approaches.

[1] *Diffusion as Shader: 3D-aware Video Diffusion for Versatile Video Generation Control*
[2] *RoPECraft: Training-Free Motion Transfer with Trajectory-Guided RoPE Optimization on Diffusion Transformers*
[3] *World-consistent Video Diffusion with Explicit 3D Modeling*
[4] *JointDiT: Enhancing RGB-Depth Joint Modeling with Diffusion Transformers*

**Questions:**

(3.2.1 Training)
How do you distinguish individual instances here? Are the instance masks generated by a segmentation model, or are they included in the dataset itself? If they are obtained from a model such as SAM during inference, mentioning this explicitly in the training section would make it clearer.

---

> ### Author Response · Authors · 2025-11-19
> **Response to Reviewer 97AZ**
>
> We thank the reviewer for the thoughtful comments and for pointing out relevant related work.
>
> ### (1) 2.5D conditioning vs full 3D
>
> We agree that full 3D conditioning, as in *Diffusion as Shader*, is powerful. Our choice of 2.5D cues is mainly driven by **practicality and accessibility**:
>
> - Full 3D typically requires multi-view supervision and accurate calibration, and often 3D reconstruction at test time.
> - Our 2.5D Instance Cues require only a single keyframe, instance masks, and monocular depth, which are much easier to obtain in typical single-view videos and editing scenarios.
>
> Importantly, our architecture does not preclude 3D: Dense RoPE and the joint head can be applied to 3D point- or mesh-based projections as well.
>
> ### (2) Dense RoPE design
>
> Dense RoPE is not intended to be a trajectory-aligned positional encoding. Our goal is to build a **learned, physics-aligned rigid-body “simulator”**: given an initial state and local motion hints, the model should *roll out* a plausible evolution of the scene, rather than copy a prescribed trajectory. For this purpose, Dense RoPE is designed to **solve token sparsity after patchify** while providing a stable anchor for the initial conditions. On the other hand, dynamic or trajectory-aware RoPE methods (e.g., RoPECraft) focus on *how* positional codes follow an external trajectory; our contribution is on *what* tokens to allocate and how to keep them anchored so that the model can learn its own physically consistent rollouts. We see these directions as complementary and will state this explicitly in the revised version.
>
> ### (3) Novelty of joint auxiliary generation
>
> We agree that using auxiliary structural signals is not new. Our contribution is in the way they are integrated:
>
> - RGB and structure share the same DiT, positional encodings, and access to Instance Cues, with only a modality embedding distinguishing them.
> - The auxiliary stream acts as a “geometry witness” that regularizes the very tokens used for RGB, rather than using a separate network or loosely coupled branch.
>
> We will extend the related work section to cover world-consistent video diffusion, JointDiT, and similar models, and clarify how our formulation differs and why it is particularly well suited for physics-sensitive control.
>
> ### (4) Instance mask generation
>
> During **training**, we use renderer-provided instance maps from the synthetic dataset to identify each object and compute per-instance flow/depth statistics. During **inference**, we obtain masks from an external segmentation model such as SAM, or from user-provided annotations when available. We will update the training and inference sections to state this clearly and avoid confusion.

---

### Official Review · Reviewer_itDA · 2025-10-30

**Soundness:** 3
**Presentation:** 2
**Contribution:** 2
**Rating:** 4
**Confidence:** 4

**Summary:**

The paper presents **STANCE**, a controllable image-to-video generation framework designed for physically coherent motion and per-instance editing, built upon the CogVideoX (5B-parameter) backbone. The approach targets the gap between intuitive per-object controls and globally consistent motion in video diffusion models.

**Key Contributions:**
1. Instance Cues: A mechanism that transforms sparse, user-provided annotations, such as object-level arrows, masks, optional mass, and a scalar ∆z into a dense, pixel-aligned 2.5D motion field used to condition the diffusion model. This formulation grounds the motion control in geometric consistency and user editability.
2. Dense RoPE: A dense control-token strategy employing rotary positional embeddings to preserve spatially localized motion anchors within a DiT-based video diffusion backbone, ensuring better control fidelity and reduced spatial drift.

Additionally, the framework supports joint generation of RGB frames and a secondary "structural witness" (e.g., depth or segmentation), intended to stabilize temporal dynamics and improve motion realism.

Quantitative evaluations using Physics-IQ and FVD demonstrate consistent improvements over strong SVD-based baselines such as SG-I2V, Drag-Anything, MoFA-Adapter, and MotionPro (all ~1.5B parameters). Qualitative results show noticeably improved motion coherence, temporal stability, and control accuracy in both synthetic and simple real-world scenes.

**Strengths:**

- Clear motivation and problem focus
    - Addresses the specific gap of achieving physically coherent, instance-controllable motion in image-to-video generation rather than general visual quality.
- Effective integration of control signals
    - Combines arrows, masks, ∆z, and optional mass into a unified 2.5D motion field, creating an interpretable and editable conditioning scheme.
- Strong experimental design
    - Ablations clearly show the contribution of each proposed component, and the method consistently outperforms all SVD-based baselines (~1.5B) in Physics-IQ and FVD despite being built on a larger 5B backbone.

**Weaknesses:**

- Model scale and fairness of comparison
    - Using a 5B-parameter CogVideoX backbone on a relatively simple synthetic dataset seems excessive. The observed gains may partly arise from sheer model capacity rather than the proposed techniques. Since all baselines (SG-I2V, Drag-Anything, MoFA-Adapter, MotionPro) are SVD-based models around 1.5B parameters, a more balanced comparison would involve using an SVD backbone or a smaller-scale CogVideo variant.
    - Additionally, instead of full fine-tuning, LoRA-based or other efficient adaptation methods could be explored, especially given the strength of the base model, to verify whether the improvements generalize without retraining the entire network.
- Computational cost of joint generation
    - The joint RGB + structure (depth/segmentation) generation likely increases training and inference cost, yet the paper doesn’t quantify the overhead or analyze the trade-off between the added cost and the relatively small metric gain.
- Limited evaluation scope
    - Experiments are confined to synthetic or very simple real scenes with short sequences and limited dynamics. The method’s effectiveness on more complex, realistic datasets remains untested.
- Physics-IQ explanation could be clearer
    - While the metric itself is cited, providing a short subsection or appendix note summarizing its components and interpretation would make the evaluation more self-contained and easier to follow for readers unfamiliar with the metric.

**Questions:**

- Computational cost
    - How much additional training time, GPU memory, or inference latency does joint RGB + structure generation introduce compared to single RGB output?
    - Is there any measurable efficiency or stability gain that justifies this added cost?
- Evaluation scope
    - The paper does not explore how STANCE behaves on longer sequences or more complex real-world scenes. Although the model is trained primarily on simple synthetic data, it would be valuable to understand its ability to generalize or adapt to more complex motion and interactions.
    - The robustness of the method to imperfect user inputs, such as noisy masks, misaligned arrows, or ambiguous object boundaries, is not analyzed. An evaluation of sensitivity to such input noise would strengthen the empirical section.
- Missing related work
    - The paper does not cite Wan-VACE, which is relevant as a recent video generation model emphasizing motion coherence and controllable conditioning.It should be discussed and cited in the related works section to provide a complete comparison landscape.
- Writing style
    - Although the paper explicitly mentions the use of language-model assistance under **"G Writing Assistance (LLM Use Disclosure)"**, some passages, especially those with repeated em-dash usage and overly polished phrasing, sound distinctly machine-generated. A careful human language revision could improve readability and overall presentation quality.

---

> ### Author Response · Authors · 2025-11-19
> **Response to Reviewer itDA**
>
> ## Response to Reviewer itDA
>
> We thank the reviewer for the careful analysis and helpful suggestions.
>
> ### (1) Model scale and fairness of comparison(comparision with cogvideoX 2B)
>
> We agree that more balanced comparisons are valuable. we have conducted experiment on a smaller version, the physics-iq score dropped by a little, demonstrating the effectiveness of our method. We have also updated our manuscript to incorporate this comparison.
>
> | Method | Physics-IQ ↑ | FVD ↓ |
> | --- | --- | --- |
> | SG-I2V | 15.42 | 113.54 |
> | Drag-Anything | 24.86 | 92.78 |
> | MoFA-Video | 29.71 | 98.30 |
> | MotionPro | 31.58 | 74.27 |
> | *VLIPP* | *36.40* | *57.90* |
> | *Ours* |  |  |
> | STANCE-2B | 45.90 | 55.97 |
> | **STANCE** | **47.62** | **50.74** |
>
> ### (2) Computational cost of joint generation
>
> For a 49-frame 512x512 sequence on a single A100, RGB-only generation takes about ~**80s**, whereas joint RGB+Depth/Seg takes about ~**120s**. Peak GPU memory and compute follow a similar moderate increase because only the video token stream is duplicated while the text and conditioning branches remain unchanged. We will report these concrete numbers in the revised version. At the same time, many efficient Transformer variants (e.g., Linformer, LongNet, Long-Short Transformer, reduce attention complexity toward linear scale. Integrating such efficient attention mechanisms into our DiT backbone is a natural piece of future work.
>
> ### (3) Physics-IQ explanation
>
> We will add a short subsection that summarizes the components of Physics-IQ and how they are combined.
>
> ### (4) Missing related work and writing style
>
> We will add Wan-VACE and related controllable video frameworks to the related work section. We also appreciate the comment on writing style and will thoroughly revise the manuscript.

---

> > ### Comment · Reviewer_itDA · 2025-11-21
> > **Post-rebuttal comment**
> >
> > Thank you for the detailed rebuttal and for adding the new experiments.
> >
> > - The new STANCE-2B results go some way toward addressing my earlier concern about backbone fairness: they suggest that the improvements are not purely an effect of moving to a 5B CogVideoX backbone, but also reflect the contribution of Instance Cues, Dense RoPE, and the joint auxiliary head.
> >
> > - The clarification on inference-time and memory overhead for joint RGB+structure generation (about a 1.5× increase for 49-frame 512×512 videos on an A100) is also helpful for understanding the runtime trade-off. One aspect I still find hard to judge is the training-time cost of the joint head. Could you give a rough comparison between RGB-only and RGB+structure training (for example, a relative wall-clock factor or GPU-day ratio at the 2B scale)? Even a coarse estimate would make the practical cost of adopting the full setup clearer.
> >
> > - Regarding scope, I appreciate that you now emphasize the focus on controllable rigid-body motion in Kubric-style scenes and simple real examples. To keep expectations aligned, I would encourage you to state explicitly that the current experiments do not cover (i) substantially longer sequences, (ii) more complex in-the-wild scenes, or (iii) systematic robustness tests for noisy masks or misaligned arrows, and to present these as directions for future work. If you already have informal observations about how performance changes under moderate noise in the masks or arrows, I would be interested to hear them, even at a qualitative level.
> >
> > - Conceptually, I see clear practical value in the combination of sparse-to-dense Instance Cues, Dense RoPE, and the joint auxiliary supervision for your target setting. If you have additional insight into typical failure modes when one of these components is removed or weakened (beyond the quantitative ablations already reported), highlighting a few concrete examples could further clarify the specific role of each part and strengthen the overall message of the paper.

---

> > > ### Author Response · Authors · 2025-11-24
> > > **comments to review itDA**
> > >
> > > Thank you again for the careful follow-up and concrete suggestions.
> > >
> > > **Training cost of the joint head.**
> > >
> > > At the 2B scale, for both configuration, we trained for 39k iterations, and observe that RGB-only training takes roughly **26 GPU-days(corresponds to 3.2 days in reality)**, while RGB+aux takes about **40 GPU-days per epoch(5 days in reality)**. We will report this relative factor in the paper so that the practical cost of adopting the full setup is clear.
> > >
> > > **Scope and limitations.**
> > >
> > > We agree it is important to set expectations explicitly. In the revised version, we will clearly state in the limitations section and will leave for future works.
> > >
> > > **Informal observations under noisy inputs.**
> > >
> > > Although we have not yet run a full noise sweep, we do have some qualitative observations:
> > >
> > > - Mild mask perturbations (small erosion/dilation or slight misalignment) tend to be handled gracefully: the object still moves in the intended direction, though contact boundaries can become slightly less clean.
> > > - For arrows, small noise usually preserves the intended motion; stronger perturbations sometimes cause the model to “average” between the noisy arrow and its learned prior, giving a weaker or slightly rotated motion.
> > >
> > > We will summarize these qualitative trends in the appendix and clearly state that a more systematic robustness study is future work.
> > >
> > > **Typical failure modes when components are weakened.**
> > >
> > > Beyond the quantitative ablations, the main qualitative failure modes we see are:
> > >
> > > - **Without Dense RoPE:** small or thin objects are the most affected. Control often works for large objects, but small ones will drift.
> > > - **Without the auxiliary structural head:**  The depth/segmentation head appears to help the model keep object boundaries and depth ordering consistent over time.
> > > - **without the RoPE duplication**: the time till convergency is longer
> > >
> > > We will add a short qualitative figure in the appendix illustrating these failure cases, to better highlight the role of each component.
> > >
> > > Lastly, if you think your concern has been addressed, please do not hesitate to raise your score.

---

> > > > ### Comment · Reviewer_itDA · 2025-11-25
> > > > **response to authors**
> > > >
> > > > Thank you for the additional details.
> > > >
> > > > The training-cost numbers at the 2B scale, the qualitative observations under noisy masks/arrows, and the description of typical failure modes when components are removed are all helpful and make the behavior and trade-offs of the method much clearer.
> > > >
> > > > I’m glad to hear that you plan to explicitly state the current scope and limitations, and to include qualitative examples illustrating failure cases and robustness trends in the appendix. I don’t have further questions at this point.

---

> > > > > ### Author Response · Authors · 2025-11-26
> > > > > **comments to reviewer itDA**
> > > > >
> > > > > Thank you again for the feedback and the follow-up. If you feel that the additional analysis has addressed your earlier concerns, we would be grateful if you could update your score.

---

### Official Review · Reviewer_kddo · 2025-11-01

**Soundness:** 3
**Presentation:** 3
**Contribution:** 3
**Rating:** 4
**Confidence:** 4

**Summary:**

This paper proposes STANCE, an image-to-video generation method that improves motion coherence and physical plausibility. The key ideas are: 1 Sparse-to-Dense motion cues: converting instance-level user inputs (arrows, masks, mass) into dense 2.5D control fields. 2. Dense RoPE: selecting active motion tokens and tagging them with rotary embeddings to prevent control collapse. 3. Joint RGB + auxiliary map generation (segmentation or depth) to stabilize spatio-temporal consistency. The method is built on CogVideoX and trained on ~200k simulated rigid-body scenes. Experiments show improvements on a Physics-IQ metric and qualitative control results.

**Strengths:**

1.Clear practical focus on controllable object motion
The design targets rigid-body interactions and physical plausibility, which is aligned with real-world applications (AR/VR, robotics, creative tools).

2.Simple modular ideas with observable gains
Sparse→dense cues and Dense-RoPE are easy to implement and demonstrate measurable improvements in control fidelity and Physics-IQ.

3.Reasonable dataset and ablation studies
The authors curated a specialized Kubric-style dataset and ablated control injection and joint auxiliary supervision.

**Weaknesses:**

1.Dataset scope is narrow and biased toward synthetic rigid-body scenes
The method is primarily validated on artificial Kubric-like collisions, which limits generalization claims. Real-world evaluations are limited to simple tabletop toys and lack diverse environments or camera motions.

2.Limited quantitative evidence and baselines
Quantitative evaluation relies mainly on Physics-IQ and FVD, without human studies or perceptual realism metrics. Baselines include SG-I2V, DragAnything, MoFA-Video, etc., but more recent state-of-the-art controllable video frameworks are missing (e.g., VACE frameworks).

3.Technical novelty is incremental
The contributions largely combine known ingredients—instance masks, flow-derived cues, and RoPE tagging—into a pipeline. While practical, the approach feels like an engineering refinement of existing image/video control pipelines. The method does not propose fundamentally new control paradigms or physical modeling insights.

**Questions:**

Does your method work for non-rigid motion as well?

---

> ### Author Response · Authors · 2025-11-19
> **Response to Reviewer kddo**
>
> We thank the reviewer for the detailed and constructive feedback.
>
> ### (1) Dataset scope and synthetic bias
>
> Our primary goal is to study **rigid-body collisions under controllable motion editing**. For this, we adopt a synthetic-first pipeline where mass, velocity, shape, and contacts can be systematically controlled.
>
> The dataset includes both:
>
> - simple rigid scenes (single/multi-object collisions), and
> - composite scenes with a large number of scanned objects and diverse environment maps,
>
> which go beyond minimal “toy” examples, although we agree they are still limited compared to fully in-the-wild data. On the other hand, collecting comparable real-world data with reliable ground truth for these quantities (e.g., object mass, true initial velocity, contact timing) at scale is extremely difficult, which is why we start from Kubric-style setups.  We will make this scope explicit and clearly state that handling more complex environments and camera motions is left for future work.
>
> ### (2) Quantitative evidence and baselines (USER STUDY)
>
> A total of 30 videos are generated, and 87 users participated in the evaluation. The number indicates the users preference in (%). We excluded SG-I2V and MOFA-video due to poor quality.
>
> | Method | Physical plausibility | Control faithfulness |
> | --- | --- | --- |
> | MotionPro | 3.7% | 11.7% |
> | VLIPP | 4.0% | 10.0% |
> | **STANCE (ours)** | **93.3%** | **78.3%** |
>
> Our baseline set includes strong controllable/video models (SG-I2V, DragAnything, MoFA-Video, MotionPro, VLIPP) that directly support drag- or mask-based video editing, which matches our setting. Recent VACE-style frameworks (e.g., Wan-VACE) mainly target text/layout or multi-modal conditioning and do not yet offer a comparable drag-based interface in our setup, so we focus on the most task-aligned baselines and will discuss VACE-type methods in the related work.
>
> ### (3) Technical novelty
>
> We respectfully disagree that the method is purely incremental. The paper contributes a coherent combination of:
>
> - a learned 2.5D **Instance Cue** interface that connects low-level flow/depth supervision to a **human- and model-friendly** control space (keyframe + instance mask + arrow + optional ∆z + mass), enabling mass and speed sweeps that change collision outcomes while keeping appearance fixed;
> - Dense RoPE is designed for a **learned, physics-aligned rigid-body simulator**, where control tokens encode *initial conditions* (who moves, where, how fast) anchored in the first frame, and the DiT learns to roll out physically plausible collisions, rather than simply following a pre-specified trajectory frame by frame as in standard trajectory-controlled motion generation, as also noted in the reviewer jb9S‘s and itDA’s strengths;
> - The auxiliary structural heads themselves are not new; our novelty lies in *how* they are integrated into the DiT backbone, which leads to faster and more stable joint convergence in practice. We thank the reviewer for pointing out this connection and will expand the related-work discussion (e.g., VideoJAM, world-consistent video diffusion) to better position our design and highlight this aspect of novelty.
>
> ### (4) Non-rigid motion
>
> Architecturally, STANCE does **not** assume rigidity: the DiT backbone and our 2.5D conditioning could model non-rigid motion as well. However, in this work, our training data and evaluation are restricted to rigid-body scenes, and our claims are focused on the physical plausibility of rigid-body collisions. We therefore do not claim performance on non-rigid phenomena such as cloth or fluids, and will make this limitation explicit while highlighting non-rigid extensions as an important direction for future work.

---

### Official Review · Reviewer_jb9S · 2025-11-02

**Soundness:** 3
**Presentation:** 3
**Contribution:** 3
**Rating:** 6
**Confidence:** 2

**Summary:**

**Summary:**
This paper presents STANCE, a controllable image-to-video generation framework that aims to improve physical and temporal coherence in diffusion-based video generation. The authors identify two key issues in prior methods—loss of signal density after encoding sparse controls and entanglement of appearance and motion supervision—and propose two simple but effective solutions: Instance Cues, which expand sparse user-editable arrows and masks into dense, camera-relative 2.5D motion fields, and Dense RoPE, which assigns spatially addressable rotary embeddings to selected motion tokens, keeping them spatially anchored during generation. The model jointly predicts RGB and an auxiliary structural map (depth or segmentation), which acts as a consistency witness. Experiments on a large synthetic dataset and real-world examples show that STANCE yields more coherent physical interactions and consistent motion, outperforming multiple baselines such as VLIPP, MoFA-Video, and MotionPro.

**Strengths:**

**Strengths:**
- Clearly identifies key weaknesses in existing controllable video generation methods: sparse tokenization and entangled training objectives.
- Proposes **Instance Cues** that make user input dense, interpretable, and camera-relative, improving control precision.
- The **Dense RoPE** mechanism elegantly addresses loss of spatial anchoring after tokenization, preserving effective motion tokens even for small or thin objects.
- Joint RGB + structural stream training is simple yet empirically effective in stabilizing geometry and improving contact plausibility.
- Comprehensive experiments, including synthetic and real-world captured videos, demonstrate consistent improvements in physical coherence (reported with Physics-IQ metric).
- The model allows intuitive editing (direction, speed, mass, ∆z) with visually consistent outcomes and realistic cause–effect relations.

**Weaknesses:**

Weaknesses:
1. While practical, the technical novelty is moderate—Instance Cues and Dense RoPE extend existing token-density and positional embedding ideas rather than introducing entirely new principles.
2. Real-world validations are limited in scale; demonstrations mostly show toy cases with one or two rigid objects, leaving uncertainty for complex multi-agent or deformable dynamics.
3. The method depends heavily on precomputed instance masks and monocular depth estimation, which may constrain general applicability outside clean lab setups.
4. It remains unclear how robust the model is when user inputs deviate from the training distributions (e.g., unrealistic mass or velocity values).
5. The auxiliary head’s role is primarily empirical; there is little theoretical or analytical discussion explaining why it particularly improves temporal stability.

**Questions:**

see the weakness

---

> ### Author Response · Authors · 2025-11-19
> **Response to Reviewer jb9S**
>
> First, we want to thank the reviewer for the insightful comments.
>
> ### (1) Technical novelty
>
> Our goal is not to introduce a new class of generative models, but to bridge **human-editable motion hints** and **token-space control** for physics-sensitive image-to-video generation.
>
> Concretely:
>
> - **Instance Cues as a 2.5D interface.**
>
>     We learn a mapping from per-instance flow and depth differences to a dense, mask-aligned 2.5D field during training, and expose it at test time as a concise interface (keyframe + mask + arrow + optional ∆z + mass). This supports systematic mass and speed sweeps that change collision outcomes while preserving appearance, which, to our knowledge, has not been studied.
>
> - **Dense RoPE for motion-token selection.**
>
>     Dense RoPE is not intended to be a **trajectory-aligned** positional encoding. Our goal is to build a **learned, physics-aligned rigid-body “simulator”**: given an initial state and local motion hints, the model should *roll out* a plausible evolution of the scene, rather than copy a prescribed trajectory. For this purpose, Dense RoPE is designed to **solve token sparsity after patchify** while providing a stable anchor for the initial conditions.
>
> - **Joint RGB in a single DiT.**
>
>     The novelty lies in **how** we integrate it into a DiT backbone. We realize joint auxiliary generation by duplicating the video stream *inside the same DiT*, with shared RoPE and shared Instance Cues, plus a lightweight modality embedding. The auxiliary stream serves as a geometry/consistency witness for the RGB stream, and leads to faster and more stable joint convergence in practice, as reflected by consistent improvements in Physics-IQ and FVD.
>
>
> We will revise the introduction and method sections to clearly present these three components as the core contributions rather than incremental tweaks.
>
> ### (2) Scale and complexity of real-world validation
>
> Our intended scope is **rigid-body dynamics** as a first step toward more general world models. To support this, we design a staged training pipeline:
>
> 1. Single Kubric-like primitives with randomized velocities to disentangle direction vs magnitude.
> 2. Multiple static/moving rigid objects to expose contact events and post-collision motion.
> 3. A large realistic subset (~100k clips) with diverse environment maps and many scanned objects.
>
> Architecturally, STANCE does not assume rigidity, but our training data and evaluation are restricted to rigid-body scenes by design. We therefore do not claim strong performance on non-rigid phenomena such as cloth or fluids. We will make this limitation explicit and highlight non-rigid extensions as future work.
>
> ### (3) Dependence on instance masks and monocular depth
>
> Our setting is controllable **video generation** from a keyframe, where high-quality instance masks (e.g., SAM) and monocular depth are standard tools. At training time we use renderer-provided instance maps; at inference we rely on SAM-like segmentation or user masks. The pipeline is modular and can benefit from future advances in segmentation and depth estimation.
>
> ### (4) Robustness to extreme mass/velocity
>
> Our Kubric dataset samples mass and velocity over a broad range, so the model is exposed to diverse combinations during training. For values within or slightly beyond this range, we observe stable and intuitive behavior (e.g., reversed collision outcome when increasing mass). For clearly unrealistic extremes, the model can produce exaggerated motion but remains stable.
>
> ### (5) Role of the auxiliary head
>
> Intuitively, the auxiliary head improves temporal stability because:
>
> - RGB and auxiliary tokens **share positional codes and attend to the same instance cue tokens**, so any inconsistency in geometry (e.g., drifting object boundaries or depth ordering) is penalized twice: once through RGB errors and once through structural errors.
> - The auxiliary loss acts as an additional ***geometric regularizer*** on the same latent tokens, discouraging solutions where RGB “cheats” by sacrificing motion coherence in favor of texture.
>
> We already see this in our ablations: adding joint RGB+Depth or RGB+Seg consistently improves Physics-IQ and FVD over RGB-only (Table 1). This interpretation is also consistent with recent joint RGB–structure/motion models such as VideoJAM (joint appearance–motion DiT) and World-consistent Video Diffusion, which similarly employ auxiliary geometry or motion heads to regularize the RGB stream and improve temporal consistency. In the revision, we will (i) explicitly cite these joint modeling works, and (ii) add a short subsection interpreting the auxiliary head as a form of multi-task, geometry-aware regularization.
>
> [1]VideoJAM: Joint Appearance-Motion Representations for Enhanced Motion Generation in Video Models (ICML 2025)
>
> [2]World-consistent Video Diffusion with Explicit 3D Modeling (CVPR 2025)

---

### Author Response · Authors · 2025-11-19

To all reviewers:
We thank you all for your insightful suggestions.

We realize that our current presentation may have given the impression that STANCE is just another trajectory-control interface on top of an I2V model. This is not our intention. Our design explicitly targets a **learned, physics-aligned rigid-body “simulator”**: Instance Cues turn low-level flow/depth into a human- and model-friendly description of initial conditions (who moves, where, how fast, with what mass), Dense RoPE keeps these conditions stably anchored in token space so the DiT can learn to roll out physically plausible collisions over time, and the jointly integrated auxiliary heads act as a geometry-aware regularizer that stabilizes this rollout. We will clarify this motivation more clearly in the revised version.

---

### Note · Program_Chairs · 2026-01-17
**Submission Desk Rejected by Program Chairs**

The following references in this submission do not refer to real documents and/or have major errors in bibliographic information:

 Jun-Yan Zhu, Jiapeng Wu, Yuxuan Shi, Tianyang Zhou, Dinghuang Yang, Joshua B Tenenbaum, Antonio Torralba, and William T Freeman. DragAnything: Interactive point-based manipulation on the generative image manifold. arXiv preprint arXiv:2306.14435, 2023.